# Epidermolysis Bullosa Acquisita—Current and Emerging Treatments

**DOI:** 10.3390/jcm12031139

**Published:** 2023-02-01

**Authors:** Deša Tešanović Perković, Zrinka Bukvić Mokos, Branka Marinović

**Affiliations:** 1Special Hospital Agram, Trnjanska Cesta 108, 10000 Zagreb, Croatia; 2Department of Dermatology and Venereology, School of Medicine, University Hospital Centre Zagreb, University of Zagreb, Kišpatićeva 12, 10000 Zagreb, Croatia

**Keywords:** epidermolysis bullosa acquisita, treatments, emerging treatments, pathogenesis, diagnostic methods

## Abstract

Epidermolysis bullosa acquisita (EBA) is a rare chronic autoimmune subepidermal blistering disease of the skin and mucous membranes, usually beginning in adulthood. EBA is induced by autoantibodies to type VII collagen, a major component of anchoring fibrils in the dermal–epidermal junction (DEJ). The binding of autoantibodies to type-VII collagen subsequently leads to the detachment of the epidermis and the formation of mucocutaneous blisters. EBA has two major clinical subtypes: the mechanobullous and inflammatory variants. The classic mechanobullous variant presentation consists of skin fragility, bullae with minimal clinical or histological inflammation, erosions in acral distribution that heal with scarring, and milia formation. The inflammatory variant is challenging to differentiate from other autoimmune bullous diseases, most commonly bullous pemphigoid (BP) but also mucous membrane pemphigoid (MMP), Brunsting–Perry pemphigoid, and linear IgA dermatosis. Due to its recalcitrance conventional treatment of epidermolysis bullosa acquisita is shown to be demanding. Here we discuss novel therapeutic strategies that have emerged and which could potentially improve the quality of life in patients with EBA.

## 1. Introduction

Epidermolysis bullosa acquisita is a rare, acquired, subepithelial bullous disorder affecting the skin, while mucous membranes are involved in 50–65% of patients [1,2,3,4]. Although clinical presentation can be similar to other autoimmune blistering diseases, EBA is distinguished by the production of antibodies against type-VII collagen [5,6,7]. This type of collagen is a major part of anchoring fibrils attaching the epidermis to the dermis. In EBA, autoantibodies destroy anchoring fibrils, leading to the epidermis’s detachment from the dermis. EBA can occur at all ages, but the most common onset peaks are the first three decades and the seventh and eighth decades of life [8,9]. Incidence of this disease is rare, between 0.08 and 0.5 per million [8,9,10,11,12,13]. According to epidemiological studies, the incidence in Republic of Korea and Germany is estimated to be higher, the highest in Germany, with 2.8 cases per million [8,14]. Gender prevalence was not noted in reported studies [8,9,10,11,12,13,14]. Patients with EBA are likely to have concomitant diseases such as rheumatoid arthritis, diabetes, cryoglobulinemia, psoriasis, and inflammatory bowel disease [15,16,17]. At the same time, an association between EBA and lymphoma was observed in 8% of patients [18]. Influence of some triggering factors, such as medications such as penicillin, vancomycin, and gentamycin, then UV-radiation and contact allergy to metals on EBA appearance have been observed [19,20,21,22].

Clinical features differ, but the two most common presentations of EBA are the classical noninflammatory–mechanobullous EBA form [23] and the inflammatory EBA form [24]. The inflammatory type of disease appears in two-thirds of reported patients. A mixed type with features of all subtypes has also been reported [24,25,26,27].

### 1.1. The Classical Form of EBA

The classical form of EBA is characterized by skin fragility, the appearance of vesicles, tense blisters, erosions on non-inflamed skin, and healing with scars and milia formation [23]. However, every region of the skin and mucous membranes can be affected, especially trauma-prone parts, such as dorsal hands, elbows, knees, feet, toes, Achilles tendon, knuckles, and sacral area [28]. Mucosal affection can be present, but it is not predominant [29]. Cicatricial alopecia and onychodystrophy may also be present. So, in differential diagnosis, porphyria cutanea tarda, and pseudoporphyria must be excluded.

### 1.2. The Inflammatory Form of EBA Has Four Clinical Presentations

#### 1.2.1. Bullous Pemphigoid-Like Variant

Patients present with tense blisters, erosions on inflamed or non-inflamed skin, crusts, and urticaria-like erythema [30]. In this variant, pruritus is commonly present. The lesions’ distribution mainly comprises the trunk, skin folds, limb extensor areas, and distal extremities [24,31], while the face can also be affected. Mucose membranes are not commonly affected in this form of EBA. Erosions heal with atrophic scars and milia cysts [29]. This form of EBA encompasses 25–50% of all cases [2].

#### 1.2.2. Mucose Membrane Pemphigoid-Like Variant

This form is characterized by the involvement of mucous membranes [32]. Mucous membranes such as oral (tongue and lips), pharynx, esophagus, epiglottis, trachea and bronchi, anal and genital mucosa, as well as conjunctiva can be partly or entirely affected [23,24,25,26,27,28,29,30,31,32,33,34,35]. Although almost half of all patients with EBA have mucosal lesions, only 5–10% have predominant mucosal affection, as in this EBA subtype [2,36]. Since this form of EBA may have severe consequences such as dysphagia, weight loss, malnutrition, and asphyxiation due to esophageal stenosis and larynx or trachea scaring [4,37,38,39,40], a multidisciplinary approach, in this case, is needed. Besides that, more aggressive treatment is needed in cases with conjunctival and laryngeal mucosa affected [32].

#### 1.2.3. Linear IgA Disease-Like Variant

This variant is characterized by tense vesicles and blisters, erythema, and urticated plaques in an annular or polycyclic form with vesicles along the edge of the lesions. This pattern is connected to linear IgA dermatosis and is termed the “string of pearls” or “crown of jewels” sign. In this variant, there is no scaring or milia. Although, this clinical cases based data are derived from observations before establishment of current diagnostic criteria mucosal involvement is reported in this variant while severe conjunctival affection is reported in only 4% of cases [36,41]. These patients can be classified as having linear IgA disease with autoantibodies against type-VII collagen or as a linear IgA disease-like variant of EBA [42,43].

#### 1.2.4. Brunsting-Perry Pemphigoid-Like Variant

In rare cases reported, the clinical presentation is characterized by subepidermal blisters, erosions, hemorrhagic crusts, and healing with pronounced atrophic scars on the head and neck. Mucosa membranes are not affected in this variant [44].

Nevertheless, accurate diagnosis of the exact EBA subtype is challenging since the clinical presentation of EBA may change from one form to another [45].

EBA is considered severe if the patient has ten or more cutaneous blisters and three or more lesions on the mucosa membranes, similar to BP [46,47] and MMP [32]. The MMP-DAI (disease activity index) score [48] is used to quantify the extent of the disease, but the cut-offs between mild, moderate, and severe stages still need to be defined.

## 2. Diagnostic Methods

Due to the variety of clinical presentations previously mentioned, the clinical picture alone is not adequate for diagnosing EBA. Widely available laboratory tests such as routine histopathology, direct immunofluorescence (DIF) microscopy, and indirect immunofluorescence (IIF) microscopy enable the diagnosis of subepithelial autoimmune bullous disease (AIBD). Still, they cannot differentiate EBA from other subepithelial AIBD [49].

Histopathology of a lesional skin or mucous membrane shows subepidermal or subepithelial cleavage; in inflammatory variants, infiltration of neutrophils with a variable number of eosinophils, monocytes, and lymphocytes in the upper dermis are seen, and in mechanobullous variant milia and fibrosis may be present [24]. DIF microscopy is performed in a perilesional skin or mucous biopsy within 1 cm of a macroscopic blister [50]. As in other diseases from the pemphigoid group, DIF will reveal linear binding of IgG and/or C3 and occasionally IgA or IgM along the dermal–epidermal junction [28]. In work done by Vodegel et al., linear deposits of IgG/IgA at the basement membrane (BMZ) show a pattern described as u-serrated [51]. The u-serrated pattern, which gives the appearance of “growing grass”, is pathognomonic for skin-bound autoantibodies against type-VII collagen found in EBA and bullous systemic lupus erythematosus (BSLE) [1,52]. All other pemphigoid diseases are characterized by an n-pattern or a “snake-like” appearance of immune deposits [52,53]. Additionally, in mucosal biopsies and the number of skin samples, the serration pattern cannot be identified. In skin samples where the serration pattern cannot be identified, a perilesional skin biopsy can be subjected to incubation with 1 mmol/L NaCl solution, which leads to cleavage within lamina lucida [54]. After separation, the detection of immune deposits in patients with EBA remains on the dermal floor, while immune deposits in patients with BP remain within the epidermal roof [49]. Since this dermal labeling is not specific, patients with anti-p200/laminin ƴ1 pemphigoid and anti-laminin 332 pemphigoid also reveal autoantibodies attached to the blister floor; there is a need for further laboratory investigation [55,56,57,58,59,60]. 

Standard indirect immunofluorescence (IIF) microscopy on normal human skin or monkey esophagus can be used for the characterization of circulating antibodies [49]. IIF on salt-splitted skin (SSS), although demanding to perform, is proven to be more sensitive than IIF on monkey or rat esophagus or unsplit skin for detecting anti-BMZ autoantibodies [55,57]. 

Fluorescence overly antigen mapping (FOAM) is a technique available in only a few laboratories worldwide where perilesional skin biopsy is subjected to co-incubation with antibodies against e.g., BP180, laminin 332, and type-VII collagen, after which co-location of autoantibodies can be detected [58,59,60,61,62]. If tissue-bound antibodies in a patient with EBA are labeled with, e.g., red fluorescence dye and staining of the biopsy with an antihuman type-VII collagen marked with, e.g., red dye, DIF will show yellowish staining along BMZ [28]. 

Additionally, anti-type-VII collagen antibodies can be detected by IIF microscopy by the usage of type-VII collagen-deficient skin from patients with dystrophic epidermolysis bullosa. IIF of an EBA patient serum on normal human skin shows linear labeling of the BMZ, while no staining is seen on type-VII collagen-deficient skin. [63,64]. Direct transmission electron microscopy (IEM) in EBA patients shows in vivo bound thick immune deposits in the anchoring fibrils (AF) zone below lamina densa (LD). At the same time, LD remains attached to the roof of a blister [65]. Enzyme-linked immunosorbent assay (ELISA) is developed to detect autoantibodies directed against collagen VII [7,66,67]. Recently developed ELISA comprises recombinant forms of desmoglein 1, desmoglein 3, envoplakin, BP180, BP230, and NC1 domain of collagen VII [68]. This multivariant assay has high sensitivity and specificity for the faster diagnosis of newly presenting AIBD in seropositive patients [68]. The novel technique used for the detection of autoantibodies to collagen VII in EBA patients, so-called IIF on BIOCHIP, is an indirect immunofluorescence method using non-collagenous 1 type-VII collagen (NC1 Col7) transfected cells on a particular slide [49]. It could be used as a substitute for ELISA in preselected cases by IIF on SSS with floor labeling due to its 50 % less cost [69,70]. Another serologic technique, immunoblotting (IB), detects antibodies to the collagen VII NC1 protein in EBA patients [71]. Although this test helps differentiate patients with anti-p200 pemphigoid who have similar findings on IIF with SSS, on the other hand, patients with inflammatory bowel disease (IBD) or BSLE who both have positive serum autoantibodies to collagen VII may be picked up [13,72,73,74]. Additionally, a group of EBA patients with only positive IgA autoantibodies may not be picked up on IB IgG assay [2].

Regarding criteria for diagnosis of EBA, consensus on one procedure that would be applicable worldwide has not been reached, but we are provided with a general framework for establishing a diagnosis of EBA that takes into account clinical presentation and available laboratory testing [49]. In an ideal scenario, a highly probable diagnosis of EBA is made by the presence of subepidermal bulla by histology, positive DIF test, and ELISA showing patients’ serum autoantibodies targeting Col7 [49]. In this scenario, no further tests are needed for EBA confirmation. The problem arises if a patient with EBA lacks autoantibodies which leads to negative IIF, SSS IIF, and ELISA tests [68,75]. In these cases, some of the additional tests and criteria need to be included; u-serration pattern by DIF microscopy with linear IgG, C3, IgA, and/or IgM deposits within epithelial BMZ, direct IEM of perilesional skin demonstrating immune deposits within AFs zone and/or lower LD and in vivo-bound immune deposits below type-IV collagen by FOAM [49]. In patients with serum-negative autoantibodies, diagnosis of EBA is considered definitive if we have a clinical bullous disorder, positive DIF microscopy, and at least one of the additional, above-mentioned criteria is satisfied [49]. Finally, suppose additional tests cannot be performed. In that case, diagnosis of EBA can be confirmed by additional dermal labeling by DIF and/or IIF on SSS and exclusion of autoimmunity against laminin 332 or the p200/laminin ƴ1 chain [49].

## 3. Treatment

Treatment of EBA includes prevention of disease progression or possible wound infection. An individualized approach in EBA treatment is essential due to the potential adverse effects that can affect present comorbidities or cause potential toxicities (Table 1). Although both inflammatory and noninflamatory form of EBA have shown challenges regarding treatment response, classical mechanobullous form have been reported to be refractory to systemic corticosteroids, azathioprine, methotrexate, and cyclophosphamide [76,77]. Cyclosporine at high doses, colchicine, high- and low-dose intravenous immunoglobulins, plasmapheresis in conjunction with immunoglobulins, and extracorporal photochemotherapy should be considered in those cases [78,79,80,81].

### 3.1. Systemic Corticosteroids

As in many other autoimmune bullous diseases, systemic corticosteroid therapy is accepted as a first choice in EBA treatment. Initial doses range from 0.5 to 2.0 mg/kg/day [82].

### 3.2. Corticosteroid Sparing Agents

Additionally, corticosteroid-sparing agents, which include colchicine, diaminodiphenyl sulfone (dapsone), methotrexate (MTX), azathioprine (AZA), cyclosporine (CSA), mycophenolate mofetil (MMF) and cyclophosphamide (CPA) have been used as a treatment option.

Colchicine is used as an adjuvant in first-line combination therapy to allow steroid tapering or as a monotherapy [2,83,84]. Initial dosages range between 0.5 and 2 mg/day with titration up to a dosage that does not cause adverse effects, which can include abdominal pain and diarrhea [85,86]. 

Similarly to colchicine, dapsone is usually used as an adjuvant therapy to systemic corticosteroids. Starting doses of dapsone are 25 to 50 mg/day and can go up to 150 mg/day during active treatment or can be used for several months as maintenance therapy in lower doses [2,87,88]. Adverse effects of dapsone therapy are hemolysis, methemoglobinemia, agranulocytosis, and peripheral neuropathy [89]. Based on the efficacy in treating BP, MTX may be a viable treatment for EBA in combination with systemic corticosteroids with or without other immunosuppressants at 20–25 mg/week [90]. However, nausea, anemia, or infection could lead to discontinuation of MTX treatment [90]. The most frequently used immunosuppressant as steroid-sparing therapy in EBA was AZA [9] but similar to MTX there are no study reports on its efficacy in EBA [91]. Adverse effects of AZA have been reported, such as fever, rash, pancreatitis, malaise, nausea, diarrhea, hepatitis, and leukopenia [92]. When using CSA mainly as adjuvant therapy, renal dysfunction was noted if used long-term or in doses higher than 5 mg/kg/day [9,93]. Hypertension, headache, tremor, paraesthesia, hypertrichosis, and hyperlipidemia have also been noted [93]. MMF, as T- and B-cell proliferation suppressor, is shown to be successful in treating EBA [94,95]. A randomized control study in pemphigus patients has demonstrated that 2 g/day offered a better risk-benefit profile than 3 g/day [96]. Some of the adverse effects could be hypertension, hyperglycemia, and infection. Clinical responses to the dose of 2–3 g/day were noted after 2 to 4 months and led to complete cessation of steroid use in those patients. However, remission with MMF maintenance dose has shown to be challenging, with some patients not being able to tap steroid usage or requiring adjuvant plasmapheresis [97,98]. Although minimal data are available, CPA is a therapeutic option only when other immunosuppressant therapy fails [9]. 

### 3.3. High-Dose Intravenous Immunoglobulin (IVIG) Therapy

IVIG has been shown effective in severe EBA cases after exhausting most available treatments of EBA. A study by Ahmed et al. showed a complete clinical response in 10 patients with a severe form of EBA, non-responsive to conventional therapy [99]. After 16 to 31 IVIG infusion therapies, after 30 to 52 months (mean 38.8 months), clinical remission was observed, with all previous treatments being tapered off over 5–9 months, allowing IVIG monotherapy [99]. Two out of ten patients reported headaches as an adverse effect [99]. After cessation of IVIG, remission was achieved in all patients throughout their follow-up, with a mean of 53.9 months [99]. A dose of 2 g/kg/cycle for three days or 400 mg/kg/day for five sequential days led to clinical improvement in most cases [99]. A recent study by Amagai et al. reported lower disease activity scores in IVIG-treated BP patients compared with placebo [100], but we are still missing a double-blind trial for EBA patients. 

### 3.4. Rituximab (RTX)

RTX has been trialed in a small number of EBA patients, and it is shown to be a promising treatment option [101]. Two established protocols are currently used: the lymphoma protocol consists of 4 weekly infusions of 375 mg/m^2^, and the rheumatoid arthritis protocol of two 1000 mg infusions separated by 14 days [102]. In a retrospective analysis, complete disease remission was achieved in 10 cases using the lymphoma protocol [9]. The efficacy of RTX was shown in randomized trials in pemphigus patients with regiment 1000 mg every two weeks on day 0 and day 14 twice and 500 mg at 12 and 18 months [103]. In a study reporting four EBA cases by Lamberts et al., treatment with 1000 mg of RTX every two weeks led to partial and chronic remission in each case and no response in two cases [104]. In a case series by Bevans et al., complete disease control was achieved in two patients with combination treatment, including dapsone, MMF, corticosteroids, and RTX [105]. They noted treatment response two to three months after RTX initiation, and while one patient could discontinue all treatments, two out of three patients required an immunosuppressant maintenance dose [105]. Combining RTX with immunoadsorption (IA) or IVIG has shown to be successful [106,107]. RTX proved beneficial in autoantibody depletion and maintenance of remission based on the Autoimmune Bullous Skin Disorder Intensity Score, as shown by Oktem et al. [107]. The latest RTX 4 cycle application report in resistant EBA by Mendes et al. showed slight improvement and stabilization of the clinical picture without achieving complete remission [108].

### 3.5. Immunoadsorption (IA)

Although plasmapheresis has been used in the treatment of pemphigus and pemphigoid, immunoadsorption (IA) took advantage over it due to selective immunoglobulin removal from circulation, no requirement of plasma components substitution, higher processing capacity per treatment, and fewer side effects such as infections and allergic reactions [109]. In several reports, combination therapy with IA and RTX in EBA shows promising potential treatment protocols for those patients [106,110,111]. In reports by Kolesnik et al. [110] and Kubisch et al. [111] where each patient was treated with IA for three consecutive days, followed by IA every week and 375 mg/m^2^ of RTX on the day after IA for four weeks, complete remission after 18 months and complete remission within 16 weeks was observed, respectively. Niedermeier et al. reported partial remission in 2 conventional therapy-resistant cases treated with two cycles of IA for four consecutive days at four-week intervals followed by 375 mg/m^2^ of RTX every week for four weeks [106]. 

### 3.6. Extracorporeal Photochemotherapy (ECP)

ECP is reported in the treatment of Sezary syndrome, mycosis fungoides, and autoimmune bullous diseases [112]. Despite the low number of published EBA patients, several reports of the use of ECP in persistent EBA showed satisfactory results, where complete remission and partial remission were noted in three cases, respectively. In contrast, one patient had no therapeutic response [79,80,113,114]. 

### 3.7. Daclizumab

A humanized monoclonal antibody against Tac antigen or CD25, daclizumab, was also reported in EBA treatment, where it showed clinical improvement in only one out of three cases [115]. 

### 3.8. Minocycline

The use of minocycline in EBA is reported in one case that was resistant to treatment with prednisolone and cyclosporine, while treatment with dapsone led to severe adverse effects [116]. A dose of 200 mg/day of minocycline prevented disease progression and allowed for the tapering of prednisolone [116]. Inhibition of granulocyte migration and cytokine production is believed to be a mechanism by which minocycline shows effectiveness in AIBD [116]. 

## 4. Pathogenesis

Studies that have advanced our understanding of the pathogenesis of EBA have been performed in various animal models of blistering disorders that lead to subepidermal or intraepidermal blistering. 

Pathophysiologic events could be divided into three phases: loss of tolerance to COL7 (induction or afferent phase), autoantibody production, and autoantibody-induced tissue damage (effector of efferent phase) [5].

### 4.1. Induction Phase

EBA susceptibility is associated with genes in and outside the major histocompatibility complex (MHC) locus. Specifically, Gammon et al. documented an association with MHC locus HLA-DR2 [117]. Since people of African descent carry the HLA-DRB1 risk allele, they are significantly more represented in a large cohort of patients with EBA [118]. From the immunization-induced EBA data, several non-MHC loci associated with specific chromosomes controlling EBA susceptibility and disease severity can be determined [5]

In addition, variations in skin blistering in antibody-induced EBA also suggest that later stages of the disease are genetically controlled [119,120]. 

In EBA patients, immunodominant regions of COL7 are recognized by autoreactive T cells [121]. For immunization-induced EBA different antigen-presenting cells that support a COL7-specific CD4 T-cell response as well as B-cells are needed. This autoantigen presentation event is present for up to 4 weeks after immunization [119]. The key role of T cells was demonstrated by Sitaru et al., who showed that mice without T cells did not develop the disease [122]. Iwata et al. reported that depletion of CD4 T cells but not CD8 T cells around the time of immunization delayed anti-COL7 autoantibodies and blister formation in mice [119]. Additionally, Hammers et al. associated an increased IFN-ƴ/IL-4 ratio in a Th1-like cytokine profile with skin blistering, while Th2-like cytokine gene expression is associated with resistance to disease induction and blister formation in mice [123].

Further research on immunoregulatory T (Tregs) and B cells (Bregs) is needed. So far, it has been shown that the autoimmunity to COL7 development is independent of Treg function [124]. However, autoantibody production was suppressed by splenic Bregs from COL-7 immunized mice [5].

As we know by now, cytokine GM-CSF and neutrophils contribute to autoantibody formation in experimental EBA [5,125], indicating contribution of B-cell helper neutrophils to T-cell independent antibody production as well as T-cell-dependent antibody response [5,125]. 

### 4.2. Autoantibody Phase

Hammers et al. found that autoantibody-producing plasma cells are more numbered in EBA liable as opposed to disease-resistant mouse strains [123]. COL7 specific plasma cells were obtained in the draining lymph nodes of immunized mice [126]. Gradual downturn trough 8–12 weeks of autoantibody titers in patients with the autoimmune bullous disease who were treated with B cell targeting immunosuppressants [127] could be justified by 4–8 week turnover time of circulating and skin-bound anti-COL7 autoantibodies in mice [128]. Challa et al. described that neonatal Fc receptors prevent autoreactive IgG degradation, which leads to tissue injury [129].

### 4.3. Effector Phase

In experimental EBA, one of the initial points of the effector phase is a complementary activation (Figure 1). Fc fragments of IgG are directly linked to autoantibody tissue damage in inflammatory EBA. If these fragments are not able to activate murine complement and link to murine FcRs blister, formation or skin lesions ex vivo in mice are found to be absent [130]. Recke et al. demonstrated that complement activation and bullae formation are dependent on human IgG1 and IgG3 anti-COL7 antibodies [120,130]. In antibody-induced EBA, Sitaru et al. found that C5-deficient mice were completely preserved from blister formation [120]. Likewise, mice lacking all neutrophil adhesion-related β2 integrins, CD18-deficient mice, and mice treated with neutrophil-depleting anti-Gr1 antibody also did not experience blister formation [131]. Additionally, expression of GM-CSF, CXCL1/2, and IL-1α/β is related to neutrophil-dependent blisters [125,132,133,134]. This shows that neutrophils contribute to T cell-independent antibody production as well as to T cell dependent antibody responses [5]. Kasperkiewicz et al. established that by activating FcγRs, FcγRIV is the only receptor required for tissue injury, while upgrading FcγRIIB opposes skin inflammation [135]. Retinoid-related orphan receptor α is proposed to be linked to signaling kinases such as P13 Kβ, Erk1/2, p39, and Akt [136,137,138]. Targeting this receptor antibody-induced EBA was partially or entirely inhibited [136,137,138].

Kopecki and Cowin found that the actin-remodeling cytoskeletal protein Flightless (Flii) affects the epidermal barrier and wound healing [139]. It has also been observed that if Flii expression is reduced in Flii +/− mice or when topical Flii-neutralizing antibodies are used, blister formation is reduced and wound healing is improved [140,141].

Tolerance loss is mediated by the interaction of APCs with autoreactive B and T cells, which leads to formation of plasma cells. Autoantibodies against COL7 are released into the blood circulation. Galactosylation of antibodies may differ. High galactosylation of IgG is crucial for anti-inflammatory properties whereas low galactosylation is pro-inflammatory. Binding of autoantibodies to DEJ in the skin induces complement deposition, pro-inflammatory cytokine and mediator release leading to leukocyte extravasation. Immune complexes bind to neutrophils and induce a signaling cascade leading to activation, including the release of ROS, chemokines and cytokines. In addition to neutrophils, other cell types are involved in split formation, such as monocytes/macrophages, NKT and γδ T cells. NKT, natural killer cell; C, complement; GM-CSF, granulocyte-macrophage colony-stimulating factor; IL, interleukin; LTB4, leukotriene B4; PDE4, phosphodiesterase 4; ROS, reactive oxygen species; APC, antigen-presenting cell; SYK, spleen tyrosine kinase; Lyn, tyrosine-Protein Kinase Lyn; HSP, heat shock protein; AKT, protein kinase B; NCF1, neutrophil cytosolic factor 1; ERK, extracellular signal-regulated kinase; HCK, tyrosine-protein kinase HCK; FGR, tyrosine-protein kinase FGR; RORα, retinoid-related orphan receptor-alpha; BLT1, leukotriene B4 receptor 1; LTB4, leukotriene B4. Data of this image are according to Frontiers in Medicine, Epidermolysis Bullosa Acquisita: The 2019 Update by Koga et al. [29].

## 5. Novel Therapeutic Implications (Table 2)

Some information can be found in Table 2.

**Table 2 jcm-12-01139-t002:** Novel therapeutic implications.

Anti-Hsp90 Treatment	Modulating induction phase
Monoclonal antibodies against CD3, CD4, IL-2R and CD 40 L	
Antigen-specific immunoadsorption	Modulating autoantibody maintenance
Saturation of neonatal Fc receptor binding sites	
Endoglycosidase (EndoS)	Modulating the effector phase
Anti-CM-CSF antibody	
IL-1 receptor blocker anakinra	
Targeting retinoid related orphan receptor α/Akt, Src, P13Kβ, Erk1/2,p38	
Soluble FcγRIIB (CD32)	
Anti-Flii antibody	

### 5.1. Modulating Induction Phase

T-cell targeting therapies such as monoclonal antibodies against CD3, CD4, IL-2R, and CD 40 L represent encouraging therapeutic approaches for EBA patients. Some of these therapies have already been successfully used in autoimmune bullous diseases [142]. 

Due to its inhibitory effects on malignant cells, a new strategy that is currently tested in clinical trials for the treatment of cancer patients uses inhibitors of the cell stress-inducible heat shock protein 90 (Hsp90) to inhibit autoreactive T cell responses [143,144,145]. As for its potent immunomodulatory action, anti-Hsp 90 treatment became a research focus for autoimmune diseases, including blistering disorders [143,145,146,147,148,149,150,151]. Anti-Hsp90 treatment inhibits autoreactive T cell proliferation [143], modulates humoral immune response at the B cell level [147], and enhances Treg function in in-vivo models of inflammation and autoimmunity [145,146]. As we stated before, IFN-γ/IL-4 is linked to the production of pathogenic autoantibodies [123]. After anti-Hsp90 treatment, reduced serum expression of IFN-γ in human Th1 cells was found [147,148]. It was concluded that promotion of clinical recovery by suppressing the production of autoantibodies and blocking the development of EBA is due to application of Hsp90 inhibitors before and after the onset of the disease. [5].

### 5.2. Modulating Autoantibody Maintenance

Immunoadsorption and plasmapheresis have been successfully used to treat AIBD, using monoclonal anti-CD20 antibodies that target B cells and high levels of circulating autoantibodies to cause depletion of both plasma cell precursors [106,128,152,153]. An advantage over today’s therapy, which is the global elimination of all B cells and antibodies, would be the use of new recombinant forms of auto-antigens [5]. Additionally, by saturating neonatal Fc receptor (FcRn) binding sites, autoantibody levels could be lowered, which represents RIV R underlining mechanisms of IVIG action [154]. Anti-FcRn treatment in experimental murine EBA have demonstrated promising results in a 4-week treatment period. Namely, 80% of the mice had improved clinical manifestation of the disease, the affected body surface area decreased by almost 50% while remission was observed in 9 of 11 mice [155].

### 5.3. Modulating the Effector Phase

A new encouraging therapeutic option could be enhancing the targeting of autoantibody binding to COL7 or inhibiting FcγRs on effector cells [5]. Established IVIG treatment is described to up-regulate FcγRIIB [145] and decrease RcγRIV expression in immunization-induced EBA [146]. Alternatively, monoclonal antibodies could also be considered due to their potential to block the binding of pathogenic autoantibodies in BP [147,148,149]. Impaired blister formation is caused by inactivation of RcγRIV and upregulation of RcγRIIB by elimination of terminal sugar residues on anti COL7 IgG by the endoglycosidase (EndoS) [150,151]. Likewise, the absence of terminal sialic acid residues caused the loss of IVIG activity [152,153]. Novel therapies for EBA patients could target the glycosylation of autoantibodies and enrich IVIG for terminal sialic acid residues [5]. Iwata et al. found that soluble FcγRIIB (CD32) successfully impaired blistering and modulated autoantibody production by blocking immune complexes binding to FcγRs [154]. By targeting cytokines GM-CSG, CXCL1, CXCL2, or IL-1, accumulation of the neutrophils in the skin can be achieved, which showed preventive and therapeutic effects in experimental EBA in mice [125,132]. Likewise, by application of an IL-1 receptor blocker, anakinra proinflammatory events in experimental EBA were counteracted [134,138]. Although the mechanism by which anti-Flii treatment impairs blister formation and improves blistered skin in experimental EBA is still to be understood, potential skin barrier treatments, for now, constitute topical application of Flii-neutralizing antibodies [140,141]. In the end, by targeting retinoid-related orphan receptor α/Akt, Src, P13Kβ, Erk1/2, p38, effector molecules related to neutrophil activation (reactive oxygen species and MMs, as well as the protein Flii) we came closer to better therapeutic results in experimental EBA [5,132,136,137,138,140,141,151]. A potential therapeutic target in autoimmune skin blistering diseases is granzyme B, which is the immune cell-secreted serine protease. Research on three independent murine models carried out by Hyrojasu et al. showed that the granzyme B-depleted model or the model where topical pharmacological inhibition was used showed a significantly reduced total blistering area compared with controls. In addition, studies show that granzyme B adds to blistering process by degrading key anchoring proteins in the dermal–epidermal junction. Further, granzyme B promotes IL-8/macrophage inflammatory protein-2 secretion, lesional neutrophil infiltration, and lesional neutrophil elastase activity. Granzyme B is found to be elevated in human pemphigoid disease blister fluids and lesional skin [156].

## 6. Conclusions

EBA is a chronic AIBD characterized by diverse clinical presentations, exacerbations, and remissions over months or years. Diagnosing EBA is challenging since many diagnostic technologies are limited to specialized laboratories. The prognosis depends on disease severity at the time of diagnosis and an adequate treatment plan. The goal of treatment is to obtain control of the disease followed by remission, which means the absence of active lesions (erythema, urticarial, bullous lesions, and erosions). A complete remission off-treatment in EBA is impossible since maintenance therapy is needed. Additionally, patients need to be familiar with the fact that cicatricial lesions are irreversible. The new knowledge of the pathogenesis of EBA and recognition of critical factors of autoimmune-mediated blister formation lead to innovative therapeutic approaches, which show promising results targeting mainly efferent EBA events in in vivo testing. Anti-Hsp90, anti-GM-CSF, and sCD32 novel treatment approaches are expected to be valuable in a classical mechanobullous EBA form where blisters are considered to form through a direct autoantibody pathway. Anti-Flii treatment could benefit both EBA variants by improving wound healing. By continuing this path, new applicable treatments may emerge in the near future, which would possibly greatly improve the living standard of patients with this rare disease.

## Figures and Tables

**Figure 1 jcm-12-01139-f001:**
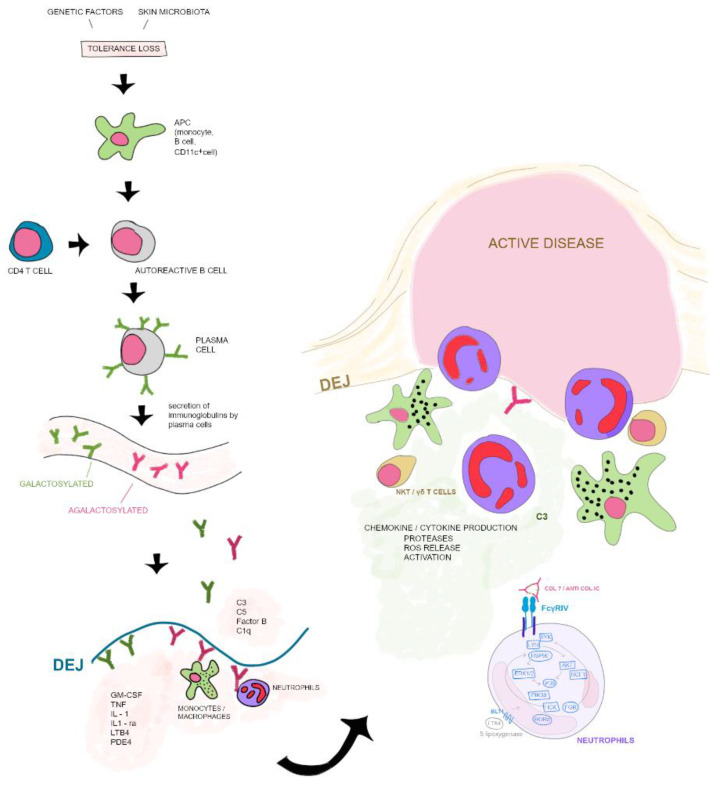
Pathogenesis pathway of EBA.

**Table 1 jcm-12-01139-t001:** EBA’s current treatment options.

1.	Systemic Corticosteroid Therapy
2.	Corticosteroid sparing agents: * 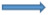 colchicine** 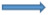 diaminodiphenyl sulfone (dapsone)** 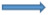 methotrexate (MTX)** 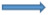 azathioprine (AZA)** 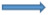 cyclosporine (CSA)** 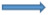 mycophenolate mofetil (MMF)** 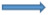 cyclophosphamide (CPA)*
3.	High-dose intravenous immunoglobulin (IVIG) therapy
4.	Rituximab (RTX)
5.	Immunoadsorption (IA)
6.	Extracorporeal photochemotherapy (ECP)
7.	Daclizumab
8.	Minocycline

## Data Availability

All the data is available from the corresponding author and can be obtained upon reasonable request.

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
