# Peer review of "Epidermolysis Bullosa Acquisita—Current and Emerging Treatments"

_jcm, 2023, doi:10.3390/jcm12031139_

Round 1

Reviewer 1 Report

xxx

Overall an excellent summary of our current understanding of EBA and potentially emerging treatments. I enjoyed reading the manuscript and have only some minor suggestions: 

- p1, l 38-43. I suggest to apply more caution as this data stems from observations before establishment of current diagnostic critera and is based on case reports / case report series only. Joust mention this fact, and all should be ok.

- An image would add to the article. It may be clinical presentation with histology and DIF, or a schematic drawing go the pathogenesis, or a schematic illustration where the mentioned drugs have their therapeutic effect. 

- under section 4.3 (or under 5.3) I would add information on the following articles that are all related to signal transduction in neutrophils: PMID: 34656615, PMID: 28576735, PMID: 36248903, PMID: 34557502, PMID: 29497423 

- likewise the impact of Granzyme B inhibition and FcRN inhibition should also be mentioned in these sections

Author Response

Dear Sir/Madam,

Thank you for your effort and help. We tried to improve this paper according to  your suggestions:

  • In rows 89, 90, 91 we mentioned that the noted data are from clinical cases observed before establishing diagnostic criteria.
  • An easy-to-understand schematic drawing of EBA pathogenesis is added under the same subtitle section.
  • In rows 515, 516, 517 the importance of neutrophils is mentioned. This is also emphasized in Figure 1.
  • Information regarding FcRN inhibition and Granenzyme B inhibition are inserted into sections 5.2 and 5.3, respectively.

Reviewer 2 Report

A comprehensive review of EBA's emergent treatments written in an easy-to-understand manner.

It would be easier to understand if there were some clinical figures of the disease.

Author Response

Dear Sir/Madam,

Thank you for your effort and help. We tried to improve this paper according to your suggestions but unfortunately, we do not have good quality, unpublished documentation of the EBA clinical presentation. Considering that it is a review article with a main focus on treatment options, we did not consider the clinical presentation “of the utmost importance” in this case. Instead, easy to understand schematic drawing of EBA pathogenesis is added under the same subtitle section.

Reviewer 3 Report

This is a very well written manuscript and comprehensive.  2 comments.

1.  What are the differences in treatment between the inflammatory and classical/mechanobullous EBA?

2.  The authors do mention potential avenues for future treatment. However the article (about 2/3) seems more a summary of the clinical presentation and current treatments of EBA.  Would an alternate title be better that suits the work of this manuscript?

Author Response

Dear Sir/Madam,

Thank you for your effort and help. We tried to improve this paper according to your suggestions and in the title, we placed „Current treatment“ as the first theme of this paper. Also, differences in treatment between the inflammatory and classical/mechanobullous EBA are inserted in section 3. under the subtitle „Treatment“.